# Assessing Neighbourhood Preference: An Evaluation of Environmental Features within Small-Scale Open Spaces

Shuyan Han [1,2], Dexuan Song [1], Feng Shi [2,3], Hu Du [4], Yuhao Zhang [5] and Mingjun Yang [6,*]

1   College of Architecture and Urban Planning, Tongji University, Shanghai 200092, China;
    hanshuyan@tongji.edu.cn (S.H.); dxsong@tongji.edu.cn (D.S.)
2   Fujian Province University Key Laboratory of Intelligent and Low-Carbon Building Technology,
    School of Architecture and Civil Engineering, Xiamen University, Xiamen 361005, China;
    shifengx@xmu.edu.cn
3   School of Architecture and Civil Engineering, Xiamen University, Xiamen 361005, China
4   School of Civil Engineering and Built Environment, Liverpool John Moores University, Cherie Booth Building,
    Byrom St., Liverpool L3 3AF, UK; h.du@ljmu.ac.uk
5   School of Architecture, South China University of Technology, Guangzhou 510006, China;
    202020104436@mail.scut.edu.cn
6   School of Architecture and Urban Planning, Shandong Jianzhu University, Jinan 250101, China
*   Correspondence: yangmingjun24@sdjzu.edu.cn

**Abstract:** Well-designed urban public spaces often attract residents and play a critical role in improving people's wellbeing. Many studies have examined the importance of one or a few environmental features in urban public spaces, such as the size of the space, greenery coverage, seating arrangements, recreational facilities, etc. However, there is a lack of systematic understanding regarding (1) which environmental features have a significant impact on the usage of urban public spaces and (2) how these features influence people's environmental preferences. To answer these questions, this investigation adopts a two-fold analytical structure: (1) first, an expert inquiry was conducted to evaluate the environmental features, and the analytic hierarchy process (AHP) was applied to determine the weight of each influencing factor; then, (2) on-site measurements were conducted across 104 spaces, accompanied by structured interviews with users of the spaces, based on which a decision tree analysis was employed to elucidate the decision-making processes of residents regarding their outdoor activities. The main findings of this investigation are as follows: (1) the site size, internal pedestrian flow, sky view factor, green-vision rate, and seat–circumference ratio are primary indicators affecting outdoor space usage, which are used in the objective evaluation index; (2) advantage value intervals for the sky view factor, green-vision rate, and seat–circumference ratio variables were calculated, and these three factors were found to significantly outweigh site size and internal pedestrian flow in terms of their effect on spatial preference. The interaction between the green-vision rate and seat–circumference ratio can affect the environmental preferences of residents: spaces with more seats exhibit lower requirements for greenery, while spaces with fewer seats should prioritise trees and greenery. Based on this study, an index based on influencing factors is proposed, enabling a better understanding of the environmental features affecting the usage of space. This study also provides valuable insights for future neighbourhood design through investigating the environmental preferences of residents, as well as the importance of various spatial features and their associated advantage value intervals.

**Keywords:** neighbourhood open space; influential feature; behaviour–environment interactions; objective evaluation index

## 1. Background

Open spaces within neighbourhoods exhibit a close connection to the daily life of citizens, serving as common venues for outdoor exercise, recreational activities, and social

interactions [1]. In contrast to public open spaces situated at a considerable distance from residences, those within a 500-m radius [2,3] offer enhanced accessibility, are more frequently utilised by residents, and hold particular significance for children, seniors [4,5], and individuals with functional limitations [6]. The proximity of these spaces to residential areas establishes them as integral components of daily life. They are closely associated with physical and leisure activities [7,8], and neighbourhood space usage has been shown to be related to health-promoting effects including both mental [9–12] and physical aspects [13–16].

Recent evidence has indicated that environmental features can greatly influence the usage of a space. This may be through fitting the needs of the users of the space, enhancing their psychological wellbeing, and improving their physiological feelings and assessments [17]. Certain features are key in promoting outdoor activity. For instance, greenery-related factors—including not only density but also quality—have been identified as key indicators affecting neighbourhood satisfaction, wellbeing, and environmental activity [18,19].

However, based on existing investigations, little is known about the behavioural effects of spatial features in the context of open spaces in neighbourhoods. Considering the diversity of outdoor circumstances and environmental attributes, it is necessary to ask which feature is most effective in affecting people's spatial preferences, and to what degree they influence the choice of space. This is important as it can address the issue of which factors should be prioritised in the design and management of open spaces, thereby promoting and encouraging the use of public open spaces in residential areas.

## 2. Literature Review

### 2.1. Comprehensive Analysis of Multiple Outdoor Influencing Factors

To detect the interactions between different types of factors and obtain a comprehensive understanding of such interactions, multiple analysis methods have been applied in various investigations, mainly focused on relatively large-scale circumstances such as comprehensive parks and whole communities rather than specific spaces based on embodied cognition. In particular, conjoint analysis has been frequently applied to study the choice-based preferences of residents regarding environmental attributes and their comparative importance [20]. Thirteen environmental attributes of neighbourhood parks were screened and, among them, nuisance, facilities (e.g., cafes and toilets), substantial greenery, light traffic, wildlife, and maintenance of the space were considered factors of high importance to elders [21,22]. Using the same statistical method, it was confirmed that friendlier communities depend on "level neighbours as friends" and quality of green spaces as key factors among the attributes of urban typology, population density, green space type, green space quality, community, and security affecting the wellbeing of residents during urban depopulation [23,24]. To gain a comprehensive understanding of environmental variables affecting the walking level within a community, principal axis factoring has been used in combination with hierarchical multiple linear regression, allowing 69 environmental variables to first be reduced to 8 and then 5 environmental factors, which were further confirmed to be significant predictors of step counts [25]. Using logistic regression analyses and principal component analysis, the level of walking and six attributes of neighbourhood open space have been analysed, and it was found that the pleasantness of an open space and lack of nuisance were associated with walking for recreation, while good paths to reach open spaces and good facilities in open spaces were conducive to the promotion of walking for transport [26].

In comparison, within small-scale public open spaces, the importance of influencing features has not yet been compared and sorted. The effects of environmental factors have been considered less comprehensively and, rather, in a relatively parallel and scattered way due to the analytical methods used in the existing literature. For example, a linear usage rate model has been applied to detect the space usage rate in a urban residential community, where environmental stimuli were compared and studied separately, and, thus, the interactions between indicators were neglected [27]. Descriptive statistical analysis

has been conducted to investigate the usage of squares, where the length and size of the space, as well as shading facilities, have been highlighted and studied [28]. To study the environmental features in parks, descriptive statistics have been used to count commonly mentioned facilities in interviews [29] and, based on *t*-tests, features including the presence of a skate park, walking paths, barbeques, picnic tables, public access toilets, lighting, and the number of trees were separately calculated [30]. In one study, the quality of design was found to be closely associated with physical activity in paediatric healing gardens using principal component analysis, while the interactions between components were not further investigated [31].

### 2.2. Investigations within Small-Scale Open Spaces

Exposure to greenery can enhance outdoor activity levels [32,33], having both psychological and physiological impacts [34]. The enclosure and greenery density in a space are closely linked to the microclimate and, thus, people's thermal comfort, further affecting their satisfaction and visiting patterns [35]. Constituent greenery factors, including plant richness, distribution, green coverage ratio, diversity of shrubs, density, vegetation quality, and maintenance, as well as greenness perception factors such as culture, serenity, and lushness, have been found to be associated with people's evaluation of a space [36] and their outdoor attendance [37–40]. The characteristics of greenery, such as whether it has more "structural" features, such as grass and trees, or more "decorative" features, such as flowers, may also change the preferences of users [41].

Space shading, which has been frequently studied together with thermal comfort and environmental adaption, can also affect the attendance of public open spaces [42]. Sunlight plays different roles in hot and cold conditions [43,44]. Shade from both shelters [45] and the tree canopy [46] are particularly preferable in hot environments. Adding shade may improve attendance in a space [47], allowing people to stay longer, conduct more static activities [48], and interact with their companions [49].

Facilities also play an important role in space usage. Seats, which are some of the most frequently used facilities in public open spaces [28], have a strong impact on the preferences of elders; in particular, the number, quality, and accessibility of seats are closely associated with the activity conducted and preference of elders [50–52]. Fitness equipment and sport courts could provide more opportunities for physical activity, especially for male adults [53]. Other equipment, such as walking and cycling paths, large lawns, barbecues and picnic equipment, play structures, and drinking fountains, have been found to play different roles in affecting the activities of specific groups, being related to the environmental preferences and behaviours of children, adolescents, and elders [22,29,54–58].

Paths and trails, which are closely associated with pedestrian flow, the accessibility of a space, and physical activity, have also been distinguished as key factors supporting the usage of neighbourhood open spaces [59].

In general, within small-scale open spaces, existing research has involved in-depth investigations on various specific factors that contribute to the design of open spaces. Meanwhile, in existing studies, the factors investigated are mostly based on a single perspective and lack the consideration of a comprehensive system. It remains difficult to answer the following questions: in addition to the impact of greening, shading, and installations, are there other factors that could have a significant impact on the usage of small-scale open spaces? Among all influencing factors, which factors play a more significant role, and do primary and secondary, subordinate, or interactive relationships exist between types of indicators? How do factors collaboratively affect public space usage, and is there a range in which environmental features make a space more attractive?

### 3. Study Area and Article Structure

#### 3.1. Aims of This Study

This study aims to examine the relative significance and behavioural effect of spatial features within small-scale open spaces in neighbourhoods. The study focused on the

following two questions: (1) Which environmental features significantly impact the public space of residential areas and should be incorporated into the analysis system? (2) How do relevant features influence environmental preferences within the analysis system?

In particular, the spaces studied are defined as open spaces that can be used by public residents, small in scale (normally less than 250 square meters), with hard paving instead of a lawn, and located either in or within a five-minute walk from a high-density residential area. The investigated spatial features are defined as those closely associated with the embodied experience of the user of the space, including facilities and morphological characteristics common to open spaces within residential areas.

As an initial exploration to provide research basics for the HENOS (Human and Environment in Neighbourhood Open Spaces) project, this study tries to gain a primary understanding of objective-based constituents and features in neighbourhood open spaces. Specifically, the aim of this study is two-fold:

- To establish a relatively comprehensive evaluation index of constituent factors affecting neighbourhood space usage, compare the weight of objective features and select indicators with significant effects;
- To analyse the decision-making mechanism of space users based on the evaluation index and assess the importance of each indicator and figure out their advantage value intervals.

Based on this, this research proposes a relatively comprehensive index system comprising influencing factors can further provide an analytical basis for future research. It also offers valuable insights for future neighbourhood design through analysing the importance of spatial features and providing their advantage value intervals.

### 3.2. Selection of Analytical Methods

Corresponding to the aims of the study, this investigation selects the analytic hierarchy process (AHP) in the initial stage to establish the relatively comprehensive index of spatial features affecting neighbourhood space usage. Then, based on the established evaluation index, preferences for environmental features are further studied using a CRT decision tree.

These two methods were selected due to the following reasons:

Firstly, the application of the AHP method can help obtain more accurate and reliable results. Compared to conventional qualitative analysis methods such as qualitative comparative analysis (QCA), which rely on subjective evaluation of a single variable at the first stage of data acquisition, or research based on grounded theory, which relies more on subjective induction by researchers in the analysis process, the AHP pairwise method comprises and compares different variables to obtain data, uses a quantitative method to calculate and check each variable's importance, and can obtain more accurate and reliable results. AHP is a widely applied decision-making approach, proposed by Saaty in 1980, which is frequently used to find the most suitable solution among various options through the evaluation of certain criteria [60]. As a hierarchical decision-making process, it reveals relationships between goals and possible alternatives [61], such as those in the fields of material selection [62] or renewable energy sources [63]. The AHP method divides a problem into different component factors, based on the overall goals to be achieved. Factors are then aggregated and combined at different levels according to their relationships, influences, and membership, forming a multi-level analytical structure model. This model ultimately boils down to the problem of determining the relatively important weight values at the lowest level (e.g., plans and measures for decision-making) relative to the highest level (overall goal). This method can be used to analyse decision-making problems when considering layered and staggered evaluation systems with target values that are difficult to quantitatively describe [64,65].

Secondly, compared to methods such as descriptive analysis and generalised linear models, the application of decision trees helps to clarify the trend of factor changes while comparing the importance of various factors and simultaneously analysing the interaction and change threshold between various factors. Specifically, the decision tree approach has

been frequently employed to search for and model the relationships, patterns, and rules existing in data sets [66]. With a structure based on a separation criterion, its process starts with a parent node, which is divided into child nodes, and ends with terminal nodes. The classification and regression tree (CRT) has been frequently adopted as an algorithm to handle continuous attributes and perform variable selection [67]. Through carrying out binary splitting of attributes, this method uses the Gini index concept as a tool to select attributes during branch classification [68].

*3.3. Article's Structure*

Corresponding to the aims and analytical methods of the study, the investigation procedure of this research is also two-fold (as shown in Figure 1):

- The initial stage of the investigation focuses on establishing a relatively comprehensive index for the evaluation of the spatial features affecting neighbourhood space usage. Four types of influencing factors, including sixteen environmental features, were sorted and calculated based on expert inquiry. The importance of each type of feature was then re-ordered and, based on the results, outstanding features were selected to form the evaluation index. Expert inquiry and AHP analysis were applied to select outstanding indicators and establish an evaluation index for spatial features affecting space usage.
- Then, based on the evaluation index established in the first stage, environmental feature preferences were further studied according to a spatial assessment and on-site measurements. Using the environmental preferences as the evaluation standard, the importance of each feature and the associated advantage value interval were further determined. Specifically, public open spaces in a neighbourhood were measured, including on-site structured interviews. Decision tree analysis was carried out to assess the importance of indicators and define the advantage ranges of indicators.

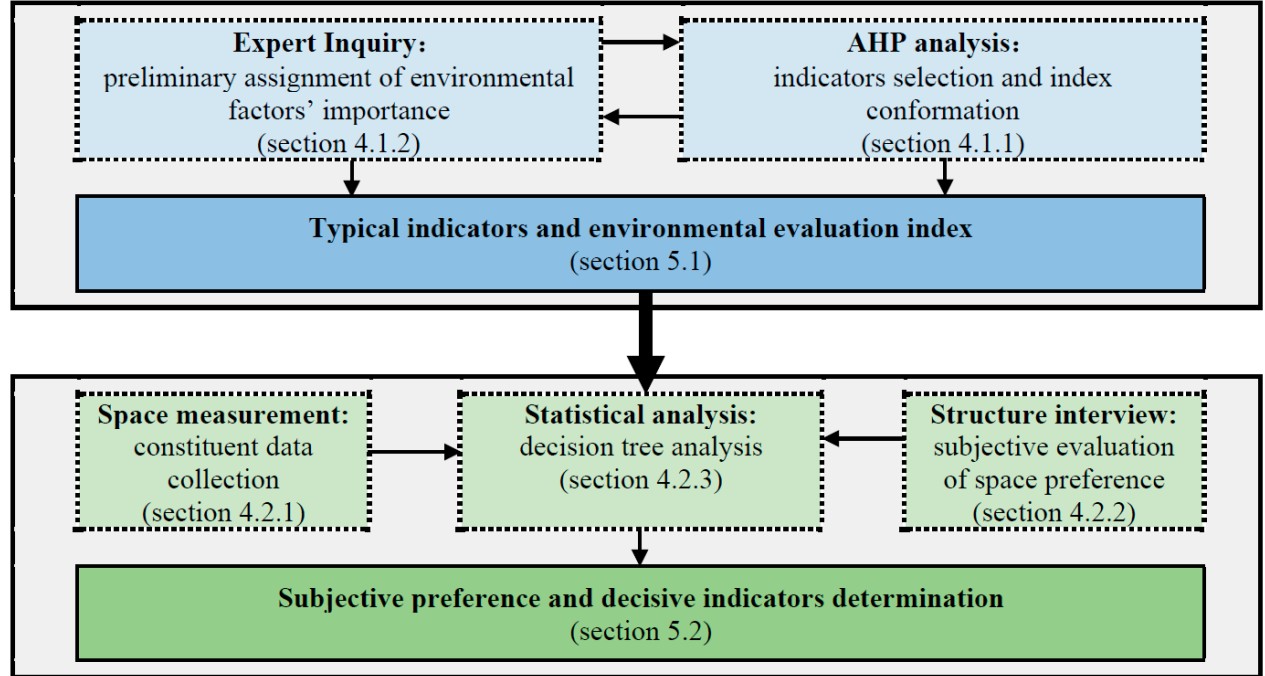

**Figure 1.** Investigation process and methods applied.

## 4. Methods

*4.1. Selection of Key Influencing Indicators and Form the Evaluation Index System*

In this section, the AHP structure is initially utilised to establish the analytical framework, which serves as the foundation for the expert questionnaire. Subsequently, through the distribution of questionnaires, experts' assessments on the spatial features are obtained in order to assign a value to this framework. Finally, based on the expert evaluations, AHP analysis is further employed to calculate the significance of different indicators and thereby select key influencing indicators and form the evaluation index system of public open space in residential areas.

### 4.1.1. AHP Analysis

To understand the contributions of the environmental features, a hierarchy was first established using the AHP structure. The structure of this system included four layers, as shown in Figure 2.

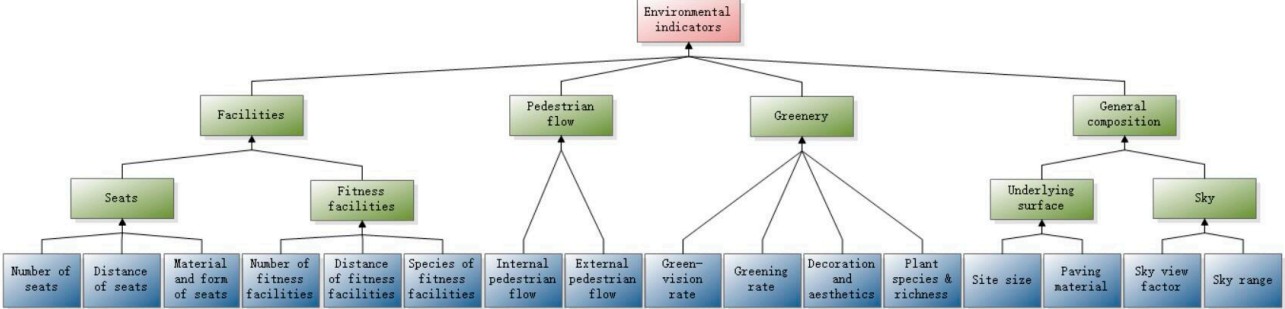

**Figure 2.** AHP analysis system.

Based on this hierarchy, the relative priorities of each criterion and sub-criterion were ranked using a numerical scale ranging from 1 to 9, in order to standardise the qualitative and quantitative performances of the priorities developed by Saaty [69]. In this stage, the ranking of factors was based on the factor evaluation scores provided by experts.

Specifically, based on investigations of small-scale open spaces, as discussed in Section 2.2, the indicators were first divided into four sub-criteria: general composition, pedestrian flow, greenery, and facilities. After that, facilities were divided into seats and fitness facilities, while the general composition was divided into the underlying surface and sky. Based on this structure, the weights of 16 types of alternatives—namely, site size, paving material, sky view factor, sky range, internal pedestrian, external pedestrian, green-vision rate, greening rate, decoration and aesthetics, plant species and richness, number of seats, distance of seats, material and form of seats, number of fitness facilities, distance of fitness facilities, and type of fitness facilities—were calculated.

Then, the contribution of each environmental constituent factor was evaluated and calculated independently, as presented in Equation (1).

Next, the value of the consistency ratio (CR) was estimated, as shown in Equations (2) and (3), where the accuracy of the calculation results is dependent on the consistency of pairwise comparison of the criteria and sub-criteria [65,70]. Therefore, any CR value greater than 0.1 indicates that the comparison matrix has inadequate consistency, such that the comparisons need to be revised and changed to reduce the inconsistency until the CR value is less than 0.1 [69].

$$A = \begin{bmatrix} a_{11} \, a_{12} \ldots \ldots a_{1n} \\ a_{21} \, a_{22} \ldots \ldots a_{2n} \\ \vdots \ldots \ldots \vdots \\ \vdots \ldots \ldots \vdots \\ a_{n1} \, a_{n2} \ldots \ldots a_{nn} \end{bmatrix} \tag{1}$$

$$I = \lambda_{max} - n/n - 1 \qquad (2)$$

$$CR = CI/RI \qquad (3)$$

In Equation (3), CI denotes the consistency index; $\lambda_{max}$ is the largest eigenvalue, n is the number of criteria, and RI is the random index, which depends on the number of criteria being compared [64,65].

4.1.2. Expert Inquiry

Questionnaires were sent to the Architecture Department of Tongji University and consultants at Shouyu Green Building Design and Consulting Co., Ltd in Shanghai, China. Experts were encouraged to give their feedback, and the respondents received gratitude and small gifts. Before the inquiry, experts' professional backgrounds were checked. All experts have been in the field of architecture design for more than 10 years and have residential design experience. None of the experts were involved in the HENOS project, and they declared no conflicts of interest and confirmed that they understood the AHP structure and comparison system before filling in the questionnaire.

After that, a total of 37 questionnaires were collected, and due to 3 incomplete questionnaires and 2 invalid responses (chose the same option for all questions, did not answer the questionnaire carefully), 5 questionnaires were considered invalid. A total of 32 experts provided valid responses (age 35–60, 18 males and 14 females), including 4 professors, 2 PhD candidates, and 26 experienced architects.

Expert inquiry was applied to assess the importance of the considered environmental features, including 4 broad headings and 16 sub-classes. In the questionnaire, with the same structure corresponding to the AHP hierarchy, spatial features were first divided into four broad headings (as shown in Figure 3), including general composition, pedestrian flow, greenery, and facilities. Then, general composition was divided into the underling surface (including the site size and paving material) and sky (including the sky view factor and sky range) sub-classes; pedestrian flow was divided into internal and external pedestrian flow; greenery was divided into green-vision rate, greening rate, decoration and aesthetics, plant species and richness; and facilities was divided into seats (including the number, distance, and material and form of seats) and fitness facilities (including the number, distance, and type of facilities).

The purpose of this survey questionnaire was to determine the relative weights between various influencing factors that affect the use of neighbourhood open spaces in residential areas. The questionnaire was designed in the form of AHP, which requires pairwise comparison of the importance between each of the influencing factors at the same level. The scale was divided into 5 levels, including values of 9, 7, 5, 3, and 1. When comparing the importance of factors on the left and right sides, 1 represents that the two factors are equally important, while 9 represents that the factor on the side where the number is located is extremely important. If two factors are between equally and extremely important, then a value of 3, 5, or 7 is chosen, based on the relative evaluation.

*4.2. Elucidation of the Decision-Making Processes of Residents Regarding Their Outdoor Activities*

Based on the established analysis system outlined in Section 5.1, this section analyses the specific impact of key indicators selected including the site size, internal pedestrian flow, sky view factor, green-vision rate, and seat–circumference ratio. Specifically, within 104 public open spaces in residential areas, five key indicators were first measured. Environmental assessment of residents was also carried out using structured interviews. A decision tree was ultimately applied to analyse the behavioural effects of indicators.

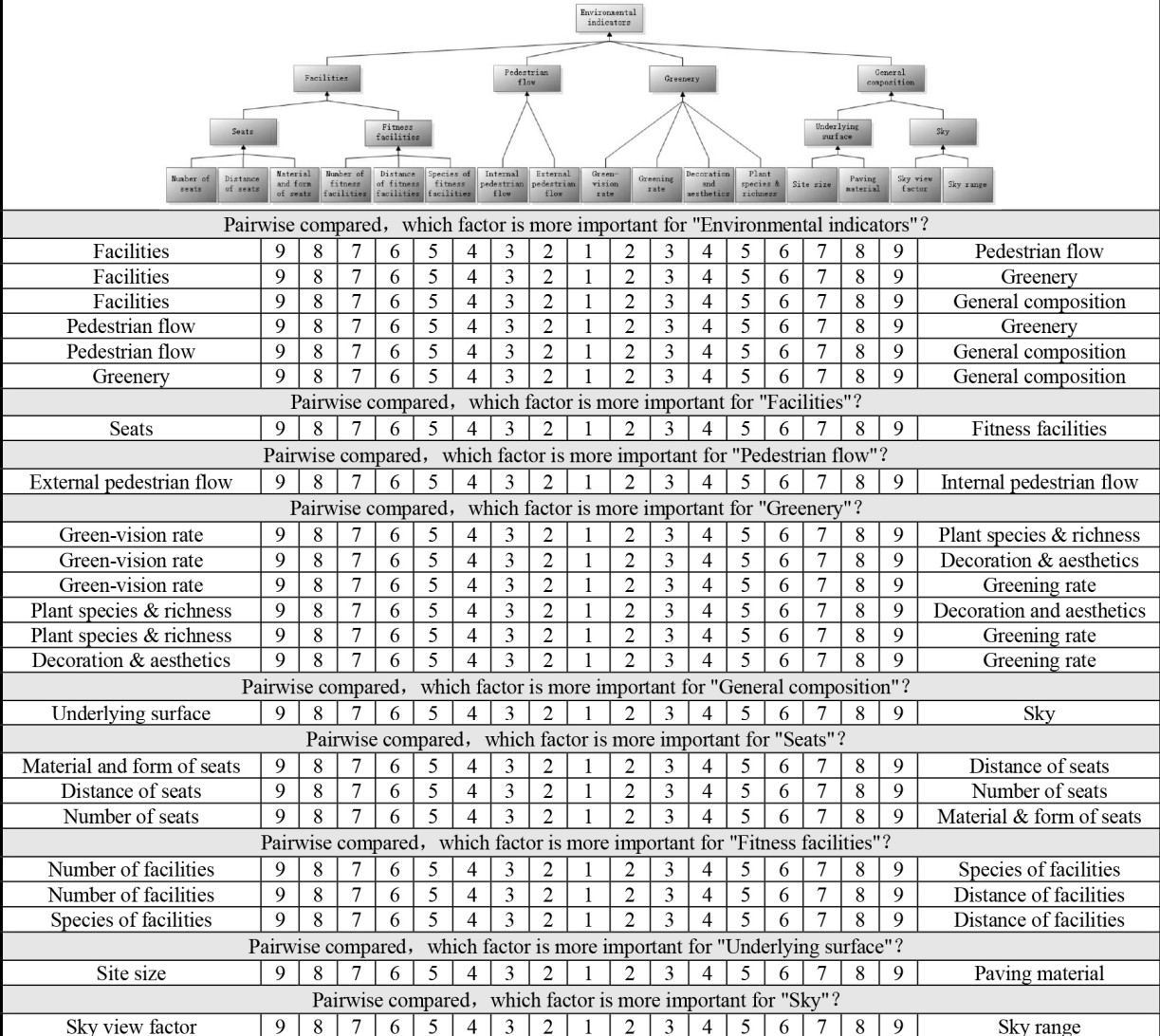

**Figure 3.** Specialist inquiry questionnaire.

### 4.2.1. Space Measurement

A total of 104 neighbourhood public open spaces (as shown in Figure 4) were measured in Yangpu District, Shanghai, China. This area has a large number of existing residential areas, built from 1950 to the present. As there was a diverse range of residential areas, the open spaces in this district were considered more representative, being conducive to case selection and providing a more comprehensive reference for neighbourhood design and renewal.

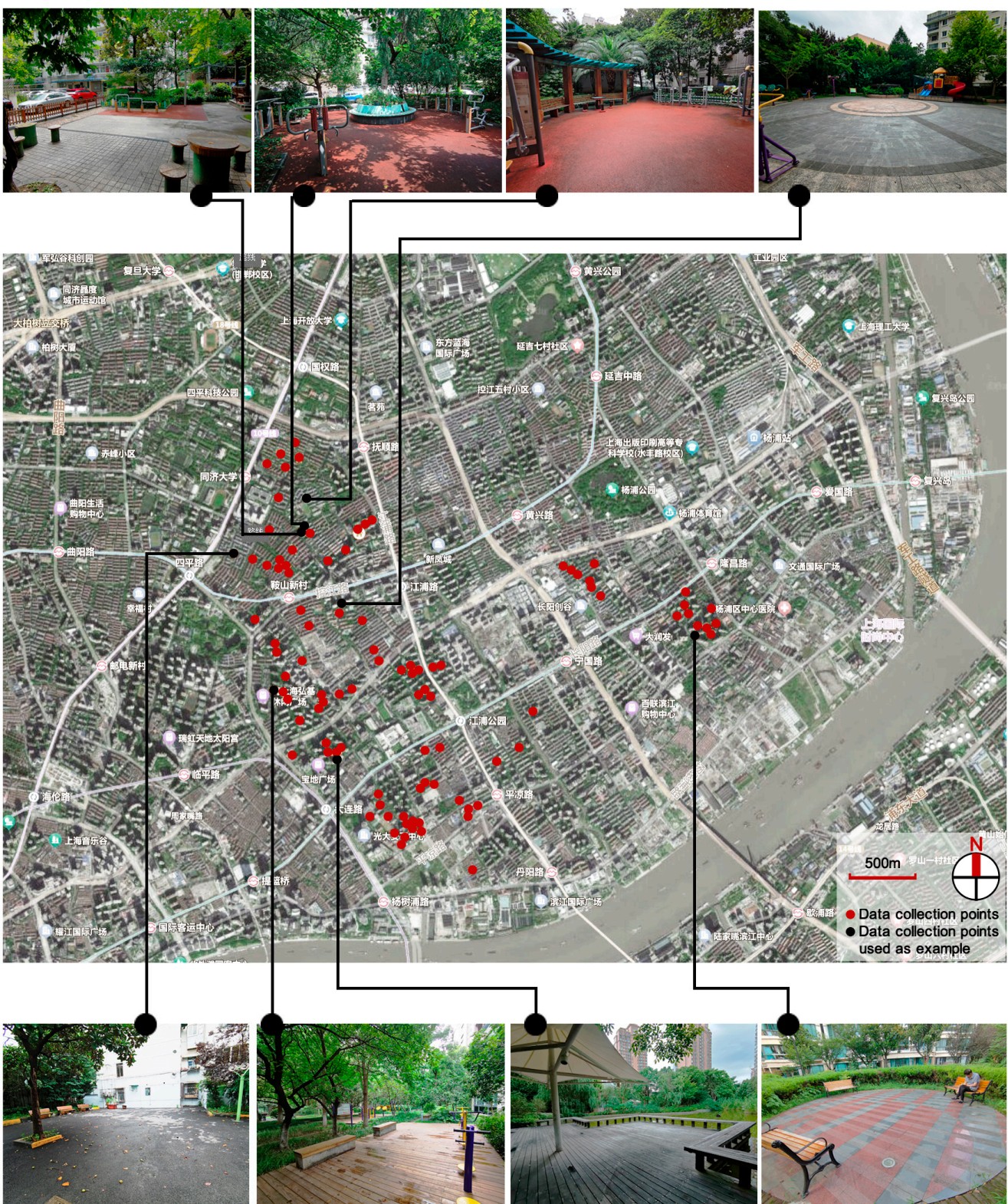

**Figure 4.** Position and examples of 104 data collection points in Yangpu District.

Based on the selected indicators (according to the analysis method described in Sections 4.1.1 and 4.1.2 and the calculation result in Section 5.1.2), five environmental features—namely, site size, internal pedestrian flow, sky view factor, green-vision rate, and seat–circumference ratio—were chosen as main indicators affecting outdoor space usage. These features and their associated measurement method are detailed in the following list:

1. Site size. The scale of a venue is directly related to activities that can be carried out and the number of people that the space can accommodate. Areas that could practically be used were measured (square metre); notably, walkways, flower terraces, and fitness equipment were excluded during measurement (as shown in Figure 5).

2. Internal pedestrian flow. This indicator is a quantitative measure of the number of people passing through the site. The more people that pass through a space, the more likely it is to attract the attention of residents; however, this also means that activities within the venue are more easily disturbed by pedestrians. The data of internal pedestrian flow were obtained through counting and grading the number of pedestrians passing through the space during non-commuting hours (9:00–11:00 a.m. and 14:30–17:00 p.m. on weekdays) within ten minutes. The grade of pedestrian flow was divided into 11 levels where, each time the number of pedestrians increased by 3, the grade increased by 1 level: when the number of pedestrians is 0–3, the grade is 0; when the number of pedestrians is 4–6, the grade is 1; when the number of pedestrians is 7–9, the grade is 2; and so on. When the number of pedestrians exceeds 30, the grade is 10.

3. Green-vision rate. The green-vision rate refers to the proportion of green vegetation within a person's field of vision. This indicator starts from the embodied scale of environmental perception and is a direct response to the greening situation of the site. For this study, the green-vision rate was calculated based on a 360-degree panoramic photo taken in the geometric centre of the space at a visual height of 1.5 m using a lens with focal length of 1.0 times (as shown in Figure 5). As the green-vision rate is an inherent indicator and the greenness of a site may differ due to seasonal changes, the green-vision rate was calculated based on photos taken in the transition season.

4. Sky view factor. This indicator refers to the proportion of the sky area to the total area when observed from the site. This indicator is related to the buildings, plants, and structures around the site. Sky view factors were captured and calculated using a fish-eye lens in the geometric centre of the field. As for the green-vision rate, this photo was also taken at a visual height of 1.5 m during the transition season (see Figure 5).

5. Seat–circumference ratio. This indicator is based on the calculation results detailed in Section 5.1.2 and is a concentrated expression of the number and distance of seats. It is calculated as the seat length divided by the total perimeter of the space, as shown in Figure 5.

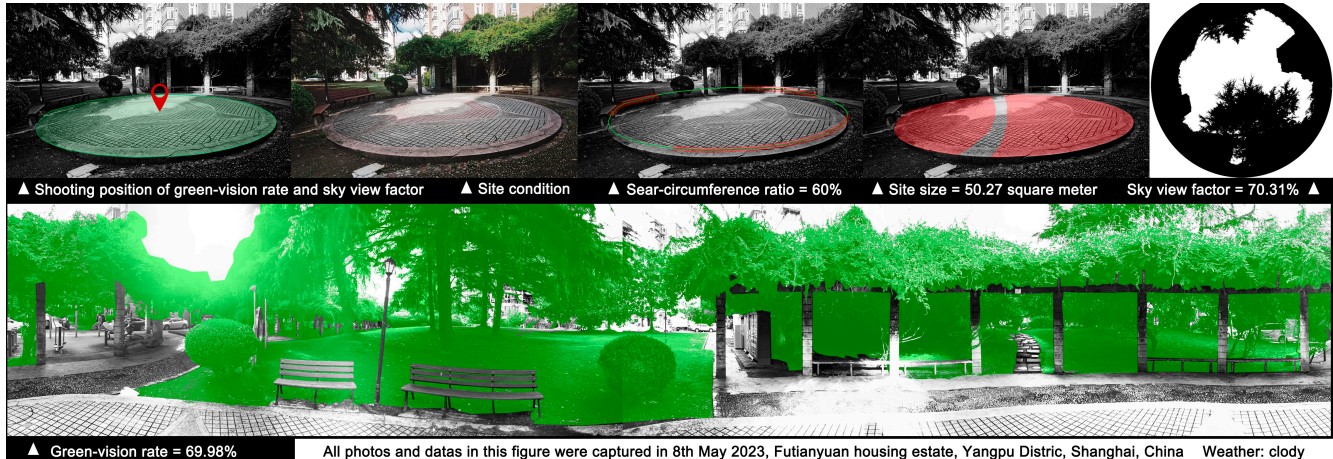

**Figure 5.** Demonstration of calculation process.

4.2.2. On-Site Structured Interview

An on-site structured interview was conducted to determine the environmental preferences of users of the spaces. For each data collection point, 28–35 habitants were inter-

viewed. For points that were close—that is, within a short distance (around 500 m) or within a 5-min walk—respondents were overlapped. A series of questions were included in this structured interview, with the aim of collecting background information, living conditions, habits of using outdoor spaces within residential areas, and spatial preferences.

Two questions were highlighted in this investigation. When provided with a picture showing the exact position of the data collection points, the respondents were first asked "have you been to the data collection points?". If their answer was yes, then respondents were further asked "do you feel this space worth frequently use in daily lives?" with a 5-scale option offered, including "definitely, probably, possibly, probably not, and definitely not". After the questionnaires were completed, factors were scored from 1 to 5 points corresponding to importance evaluation by the respondents.

### 4.2.3. Decision Tree Analysis

Average assessment of spaces was set as the dependent variable, while the site size, internal pedestrian flow, sky view factor, green-vision rate, and seat–circumference ratio were set as independent variables. Based on the sample size, the minimum number of cases in the parent node was set as 10, the minimum number of cases in a child node was set as 5, and the maximum depth of the decision tree was set to 5 using SPSS 25.0.0.0 software.

## 5. Results

### 5.1. Influential Indicators and Environmental Evaluation Index

Based on the expert inquiry and AHP analysis, this section aims to identify the main influencing features affecting neighbourhood public open space usage, with the aim to establish a relatively comprehensive evaluation index system.

### 5.1.1. Expert Indicator Evaluation

After collection, re-organisation, and averaging of the questionnaires, the environmental indicator judgment matrices of the 32 experts were combined, as shown in Table 1. In this table, each sub-factor's importance in contributing to the target indicators was compared, based on which the initial weights ($W_i$) were calculated. According to the hierarchy structure and expert inquiry form (as shown in Figures 2 and 3), the calculation results for the three layers were as follows:

1. The contribution to overall environmental indicators $W_i$ was assigned a value of 1, with the $W_i$ for the four factors of facilities, pedestrian flow, greenery, and general composition being 0.2596, 0.1202, 0.1707, and 0.4495, respectively. The general composition was considered to be the most influential factor, followed by facilities, greenery, and pedestrian flow.

2. As a subordinate of facilities ($W_i$ = 1.0000), seats ($W_i$ = 0.8750) presented an overwhelming advantage when compared with fitness facilities ($W_i$ = 0.1250). In terms of their contribution to pedestrian flow ($W_i$ = 1.0000), internal ($W_i$ = 0.8571) and external pedestrian flow ($W_i$ = 0.1429) were included, and internal pedestrian flow was considered to be more influential. In the greenery category ($W_i$ = 1.0000), the impact of the green-vision rate ($W_i$ = 0.6714) was far above those of the greening rate ($W_i$ = 0.1998), decoration and aesthetics ($W_i$ = 0.0634), and plant species and richness ($W_i$ = 0.0654). Considering the environmental general composition ($W_i$ = 1.0000), the impact of the underlying surface ($W_i$ = 0.8333) was higher than that of sky ($W_i$ = 0.1667).

3. Among all seat characteristics ($W_i$ = 1.0000), the impacts of the number of seats ($W_i$ = 0.5621) and distance ($W_i$ = 0.3748) were more significant than the impact of material and form ($W_i$ = 0.0632). Regarding the aspect of fitness facilities ($W_i$ = 1.0000), distance ($W_i$ = 0.6551) was more important than the type ($W_i$ = 0.2114) or number of facilities ($W_i$ = 0.1335). In addition, contributing to the underlying surface ($W_i$ = 1.0000), the site size indicator ($W_i$ = 0.9000) was more influential than the paving material ($W_i$ = 0.1000). Finally, the sky view factor ($W_i$ = 0.8750) presented an overwhelming advantage when compared with the range of sky ($W_i$ = 0.1250).

**Table 1.** Judgment matrix of environmental indicators.

| Contribute to: Environmental Indicators | Facilities | Pedestrian Flow | Greenery | General Composition | Wi |
|---|---|---|---|---|---|
| Facilities | 1.0000 | 2.0000 | 2.0000 | 0.5000 | 0.2596 |
| Pedestrian flow | 0.5000 | 1.0000 | 0.5000 | 0.3333 | 0.1202 |
| Greenery | 0.5000 | 2.0000 | 1.0000 | 0.3333 | 0.1707 |
| General composition | 2.0000 | 3.0000 | 3.0000 | 1.0000 | 0.4495 |
| **Contribute to: Facilities** | Seats | Fitness facilities | | | Wi |
| Seats | 1.0000 | 7.0000 | | | 0.8750 |
| Fitness facilities | 0.1429 | 1.0000 | | | 0.1250 |
| **Contribute to: Pedestrian flow** | Internal pedestrian flow | External pedestrian flow | | | Wi |
| Internal pedestrian flow | 1.0000 | 6.0000 | | | 0.8571 |
| External pedestrian flow | 0.1667 | 1.0000 | | | 0.1429 |
| **Contribute to: Greenery** | Green-vision rate | Greening rate | Decoration and aesthetics | Plant species and richness | Wi |
| Green-vision rate | 1.0000 | 6.0000 | 8.0000 | 9.0000 | 0.6714 |
| Greening rate | 0.1667 | 1.0000 | 5.0000 | 3.0000 | 0.1998 |
| Decoration and aesthetics | 0.1250 | 0.2000 | 1.0000 | 1.0000 | 0.0634 |
| Plant species and richness | 0.1111 | 0.3333 | 1.0000 | 1.0000 | 0.0654 |
| **Contribute to: General composition** | Underlying surface | Sky | | | Wi |
| Underlying surface | 1.0000 | 5.0000 | | | 0.8333 |
| Sky | 0.2000 | 1.0000 | | | 0.1667 |
| **Contribute to: Seats** | Number of seats | Distance of seats | Material and form of seats | | Wi |
| Number of seats | 1.0000 | 2.0000 | 7.0000 | | 0.5621 |
| Distance of seats | 0.5000 | 1.0000 | 8.0000 | | 0.3748 |
| Material and form of seats | 0.1429 | 0.1250 | 1.0000 | | 0.0632 |
| **Contribute to: Fitness facilities** | Number of fitness facilities | Distance of fitness facilities | Species of fitness facilities | | Wi |
| Number of fitness facilities | 1.0000 | 0.2500 | 0.5000 | | 0.1335 |
| Distance of fitness facilities | 4.0000 | 1.0000 | 4.0000 | | 0.6551 |
| Species of fitness facilities | 2.0000 | 0.2500 | 1.0000 | | 0.2114 |
| **Contribute to: Underlying surface** | Site size | Paving material | | | Wi |
| Site size | 1.0000 | 9.0000 | | | 0.9000 |
| Paving material | 0.1111 | 1.0000 | | | 0.1000 |
| **Contribute to: Sky** | Sky view factor | Sky range | | | Wi |
| Sky view factor | 1.0000 | 7.0000 | | | 0.8750 |
| Sky range | 0.1429 | 1.0000 | | | 0.1250 |

5.1.2. Weight Calculation and Environmental Indicator Selection

After the construction of the judgment matrix, the CR values were applied to verify the rationality of the hierarchy process. If the CR value is less than 0.1, then the evaluation system is considered to be reasonable and can be adopted. In this matrix, as shown in Table 2, the CR values of general composition (CR = 0.0268), pedestrian flow (CR = 0.0000), greenery (CR = 0.0618), facilities (CR = 0.0000), underling surface (CR = 0.0000), sky (CR = 0.0000), seats (CR = 0.0741), fitness facilities (CR = 0.0520) are all less than 0.1, which means this hierarchy is acceptable.

Based on the judgment matrix, the weight and rank of each indicator can be calculated. Specifically, the ranking of 16 indicators is as follows: site size (weight = 0.3371) > number of seats (weight = 0.1277) > green-vision rate (weight = 0.1146) > internal pedestrian flow (weight = 1030) > distance of seats (weight = 0.0851) > sky view factor (weight = 0.0656) > paving material (weight = 0.0375) > greening rate (weight = 0.0341) > distance fitness facilities (weight = 0.02130) > external pedestrian flow (weight = 0.0172) > material and form of seats (weight = 0.0143) > plant species and richness (weight = 0.0112) > decoration

and aesthetics (weight = 0.0108) > sky range (weight = 0.0094) > species of fitness facilities (weight = 0.0069) > number fitness facilities (weight = 0.0043).

**Table 2.** Weight, consistency ratios, and rank calculated by AHP.

| Criteria | CR | Weight | Sub-Criteria | CR | Weight | Alternatives | Weight | Rank |
|---|---|---|---|---|---|---|---|---|
| General composition | 0.0268 | 0.4495 | Underling surface | 0.0000 | 0.3746 | Site size | 0.3371 | 1 |
| | | | | | | Paving material | 0.0375 | 7 |
| | | | Sky | 0.0000 | 0.0749 | Sky view factor | 0.0656 | 6 |
| | | | | | | Sky range | 0.0094 | 14 |
| Pedestrian flow | 0.0000 | 0.1202 | | | | Internal pedestrian flow | 0.1030 | 4 |
| | | | | | | External pedestrian flow | 0.0172 | 10 |
| Greenery | 0.0618 | 0.1707 | | | | Green-vision rate | 0.1146 | 3 |
| | | | | | | Greening rate | 0.0341 | 8 |
| | | | | | | Decoration and aesthetics | 0.0108 | 13 |
| | | | | | | Plant species and richness | 0.0112 | 12 |
| Facilities | 0.0000 | 0.2596 | Seats | 0.0741 | 0.2272 | Number of seats | 0.1277 | 2 |
| | | | | | | Distance of seats | 0.0851 | 5 |
| | | | | | | Material and form of seats | 0.0143 | 11 |
| | | | Fitness facilities | 0.0520 | 0.0325 | Number fitness facilities | 0.0043 | 16 |
| | | | | | | Distance fitness facilities | 0.0213 | 9 |
| | | | | | | Species of fitness facilities | 0.0069 | 15 |

In all factors, the weight of site size, number of seats, green-vision rate, internal pedestrian flow, distance of seats, and sky view factor demonstrate significant difference compared with other factors (weight > 0.005). Therefore, these factors were selected as environmental constituent evaluation indicators. Among them, the number of seats and distance of seats were merged and adjusted to the seat–circumference ratio due to the consistency of evaluation objects. This adjustment can help to unify the processing and calculation of seat-related indicators and provide a more comprehensive evaluation of the overall composition of seats.

Overall, based on weight calculation and evaluation factor adjustment, five indicators including the site size, internal pedestrian flow, sky view factor, green-vision rate, and seat–circumference ratio were chosen as the main indicators affecting outdoor space usage in the neighbourhood objective evaluation index.

### 5.2. Subjective Preference Based on Environmental Evaluation Index System

Based on the objective evaluation index system including the five indicators (site size, internal pedestrian flow, sky view factor, green-vision rate, and seat–circumference ratio) selected in Section 5.1, this section aims to determine how these indicators affect the outdoor preferences of residents. Through spatial measurements, on-site structured interviews, and using the decision tree statistical analysis method, this section examines how residents make outdoor decisions.

#### 5.2.1. Distribution of Environmental Features

Based on the site size, internal pedestrian flow, sky view factor, green-vision rate, and seat–circumference ratio measurements obtained for the 104 public open spaces in Yangpu District, Shanghai, China, the distributions of these spatial features are shown in Figure 6, and detailed in the following list:

1.   Site size. Regarding the aspect of space size, ranging from 1 to 220 square meters, a large difference was observed among the 104 data collection points. The size of most spaces was concentrated between 0 and 25 square meters, a few spaces were in the range of 25–50 square meters, and the number of spaces larger than 50 square meters was relatively small. The mean value of size for all spaces was 30.53, while the median

value was only 13.25; this is mainly due to the existence of a few large spaces in the residential areas.

2. Seat–circumference ratio. The distribution of the seat–circumference ratio was relatively scattered. Some spaces did not have seats, while some had many seats all around the space. As the seat–circumference ratio increases, the number of spaces shows a general downward trend. The mean and median values of the seat–circumference ratio are relatively close, being 29.01 and 22.75 square meters, respectively.

3. Internal pedestrian flow. This was calculated based on the number of people passing through the space within ten minutes, resulting in a value between grade 0 and grade 10. There were significant discrepancies between different spaces, and the mean and median of internal pedestrian flow values were 3.71 and 3.00, respectively.

4. Green-vision rate. Compared with the other parameters, the distribution of the green-vision rate was relatively concentrated in 104 measured open spaces, being mainly distributed within the range of 25% to 75%. The highest green-vision rate of a space was 77.03%, while the lowest green-vision rate was only 1.85%. The median and average values of this parameter were very close, being 44.39% and 45.92%, respectively.

5. Sky view factor. Due to the high-density construction in Shanghai, the shading rate of neighbourhood open spaces is at a relatively high level. Based on the 104 open spaces measured, the sky view factor was concentrated between 34.22% and 99.95%, and most spaces had a value in the range of 70–95%.

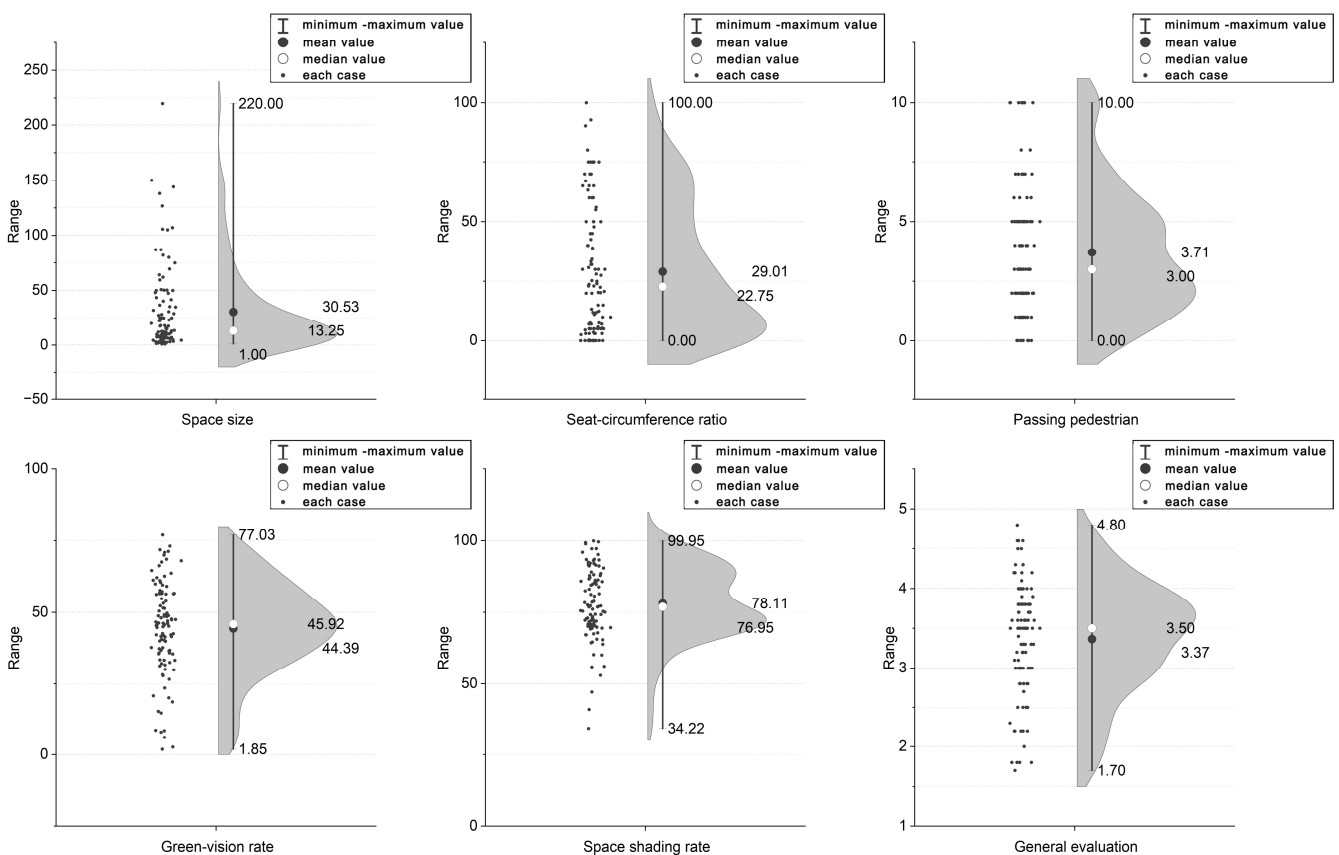

**Figure 6.** Data distribution.

### 5.2.2. Distribution of Environmental Subjective Preference

Environmental preference was evaluated using a five-point scale, as shown in Figure 6. The most preferred space was scored 4.80, while the least preferred space was scored 1.70. There were relatively more spaces with scores within the range of 3.00–4.00, and the preference values of the sites presented a peak in the range of 3.70–3.80, with the mean and

median values for neighbourhood spaces being 3.37 and 3.50, respectively. Overall, the assessment of open spaces in neighbourhoods was relatively moderate, with only a few spaces evaluated as extremely preferred or unliked.

### 5.2.3. Indicators Affecting Spatial Attendance

As shown in Figure 7, a decision tree was created using the CRT algorithm, with a structure based on the separation criterion detailed in Section 4.2.3. The tree was divided into 12 child nodes, ending with 7 terminal nodes. Based on this structure, the decision-making methods (as shown in Figure 8) can be summarised as follows:

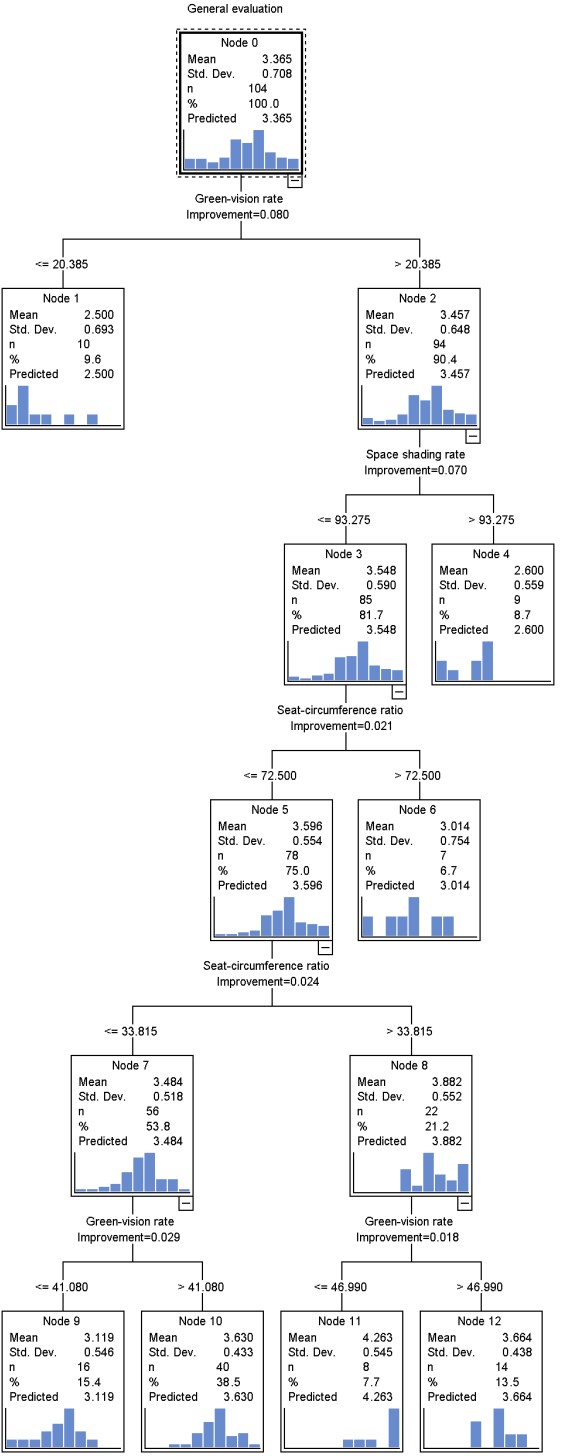

**Figure 7.** Structure of decision tree.

| BRANCH 1 (Terminal node 1) | IF: | Green-vision rate of the space <= 20.385% |
|---|---|---|
| | THEN: | The predict preference mark of the space is 2.500 |
| BRANCH 2 (Terminal node 4) | IF: | Green-vision rate of the space > 20. 385% |
| | AND: | The shading rate of the space > 93.275% |
| | THEN: | The predict preference mark of the space is 2.600 |
| BRANCH 3 (Terminal node 6) | IF: | Green-vision rate of the space > 20. 385% |
| | AND: | The shading rate of the space <= 93.275% |
| | AND: | Seat-circumference ratio of the space > 72.500 |
| | THEN: | The predict preference mark of the space is 3.014 |
| BRANCH 4 (Terminal node 9) | IF: | Green-vision rate of the space is between 20. 385 and 41.080% |
| | AND: | The shading rate of the space <= 93.275% |
| | AND: | Seat-circumference ratio of the space <= 33.815% |
| | THEN: | The predict preference mark of the space is 3.119 |
| BRANCH 5 (Terminal node 10) | IF: | Green-vision rate of the space >41.080% |
| | AND: | The shading rate of the space <= 93.275% |
| | AND: | Seat-circumference ratio of the space <= 33.815% |
| | THEN: | The predict preference mark of the space is 3.630 |
| BRANCH 6 (Terminal node 11) | IF: | Green-vision rate of the space is between 20. 385 and 46.990% |
| | AND: | The shading rate of the space <= 93.275% |
| | AND: | Seat-circumference ratio of the space is between 33.815 and 72.500% |
| | THEN: | The predict preference mark of the space is 4.263 |
| BRANCH 7 (Terminal node 12) | IF: | Green-vision rate of the space > 46.990% |
| | AND: | The shading rate of the space <= 93.275% |
| | AND: | Seat-circumference ratio of the space is between 33.815 and 72.500% |
| | THEN: | The predict preference mark of the space is 3.664 |

**Figure 8.** Branches of decision tree.

1. Highest quality: Spaces with a green-vision rate between 20.385% and 46.990%, shading rate less than 93.275%, and seat–circumference ratio between 33.815% and 72.500% were the most popular and highly evaluated (predicted score: 4.263).

2. Good quality: When the shading rate is less than 93.275%, two types of spaces are more preferred: when the seat–circumference ratio of the space is less than 33.815% and the green-vision rate is higher than 41.080%, the predicted evaluation score of the space is 3.630; when seat–circumference ratio of the space is between 33.815% and 72.500% and the green-vision rate is higher than 46.990%, the predicted evaluation score of the space is 3.664.

3. Unacceptable: When the green-vision rate of the space is less than 20.385% or the shading rate of the space is higher than 93.275%, the space will be considered least worth visiting, with predicted evaluation scores of 2.500 and 2.600, respectively.

Comparing the five selected factors (i.e., site size, internal pedestrian flow, sky view factor, green-vision rate, and seat–circumference ratio) initially set as independent variables, the constructed decision tree structure only included three of them, while both site size and internal pedestrian flow were excluded. Based on this result, and considering the importance of indicators shown in Figure 9, the site size and internal pedestrian flow did not have a significant impact; in comparison, the sky view factor, green-vision rate, and seat–circumference ratio can be considered to be the main factors affecting subjective behavioural decision-making.

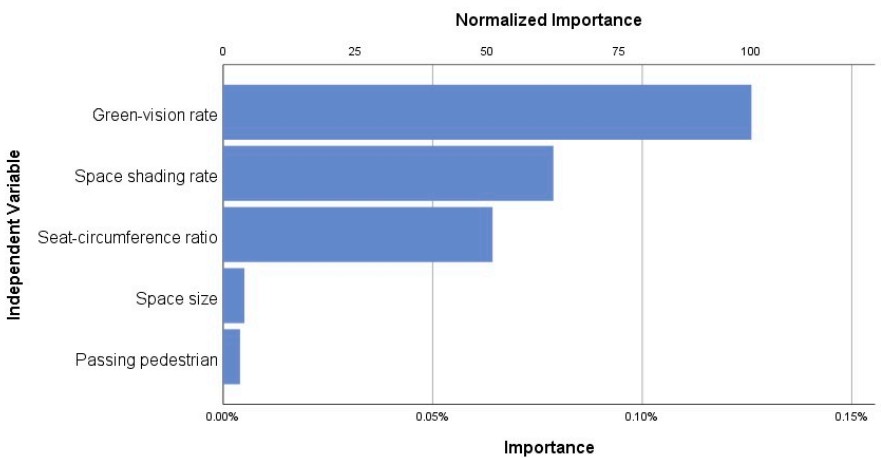

**Figure 9.** Indicator importance in decision tree.

1. Green-vision rate

The green-vision rate was the most influential feature affecting preference for a space. When this indicator was below 20.385%, the preference for open spaces was the lowest (predicted score: 2.500) among all branches in the decision tree. Thus, whether the green-vision rate is below 20.385% is the primary factor affecting neighbourhood open space preference.

In addition, the preference for the green-vision rate is influenced by the distribution of seats (as shown in Figure 10). When the shading rate of a space is higher than 93.275% and the seat–circumference ratio is lower than 72.5%, the preferred green-vision rate will change under a difference in the seat–circumference ratio. For spaces with seat–circumference ratio less than 33.815%, sites with green-vision rate higher than 41.080% are preferred; when the seat–circumference ratio is between 33.815% and 72.500%, the preferable green-vision rate will be in the range of 20.385% to 46.990%. This indicates an interaction relationship between the green-vision rate and seat–circumference ratio, and the arrangement of seats can affect the cognition of greenery by users of the space: spaces with more seats have a lower requirement for greenery, while, for sites with fewer seats, more trees and greenery will be needed.

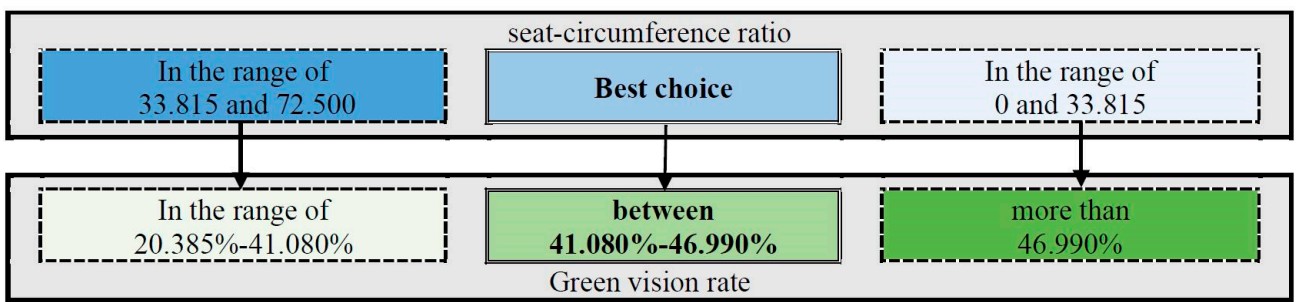

**Figure 10.** Interaction between seat-circumference ratio and green-vision rate.

In general, a green-vision rate higher than 20.385% is a necessity for highly rated spaces. When the spatial quality requirements are further improved, it is necessary to sub-divide this indicator due to the influence of the sky view factor and seat–circumference ratio. Considering the overall distribution of this feature, the assessment of the space is extremely low when the green-vision rate is below 20.385, while the space will be preferred in all situations when it is in the range of 41.080% to 46.990%; besides these two ranges, the green-vision rate must be adjusted according to the seat–circumference ratio.

2. Sky view factor

Regarding the sky view factor, the shading rate of 93.275% is a boundary. With a space shading rate greater than 93.275%, the predicted preference score of the space is 2.600, while the predicted score increases to more than 3.014 when this factor is less than or equal to 93.275%. If an open space requires a higher preference score, a space shading rate of less than 93.275% is necessary.

3. Seat–circumference ratio

Although the impact of the seat–circumference ratio was not found to be decisive, interval differences were still observed for this parameter. Spaces with a seat–circumference ratio in the range of 33.815% and 72.500% are preferred (predicted score: 3.882) over spaces with fewer seats (predicted score: 3.484). At the same time, when spaces have too many seats with a seat–circumference ratio higher than 72.500%, the space preference of residents will drop sharply (predicted score: 3.014).

## 6. Discussion

### 6.1. Importance of Environmental Greening Has Been Reconfirmed

In existing investigations, a strong association has been found between greenness and physical activity [3]; moreover, in small-scale spaces, greenery may directly affect the amount of activity conducted. The importance of eye-level greenery has been previously emphasised, such as in the case of streets, where the impact of eye-level greenery was associated with total activity time [71]. Previous investigations focused on larger-scale environments have found that levels of physical activity were higher in greener neighbourhoods [15].

This study confirmed the importance of greenery from the perspective of embodied perception, as eye-level greenery was found to play a more significant role when compared with other environmental indicators, including the site size, internal pedestrian flow, sky view factor, and seat–circumference ratio.

In addition, this study further specified the effect of eye-level greenery, as it was found that the advantage value of green-vision rate is in the range of 41.080–46.990%; furthermore, when the value of this indicator is lower than 20.385%, the spatial assessment will significantly decrease.

### 6.2. Space Shading: Not Only Linked with Thermal Comfort

This study revealed the negative effects of a low sky view factor: for spaces with shading rates higher than 93.275%, the general environmental preference showed a sharp decline.

The influences of shading and sky view factors have been specified in many past investigations, most of which were based on thermal comfort, among which the influences of specific seasons and real-time climate have been emphasised [43,45]. On the one hand, this type of investigation was more focused on the thermal feeling, providing a more detailed description of the effects of temperature and solar radiation. On the other hand, whether shading changes environmental cognition and how to evaluate the effects of shading from a visual perspective are still worth investigating. Studies conducted in Taiwan and Rome have shown that spaces with a lower sky view factor [72] and higher shading rate [73] will be more highly preferred.

### 6.3. Seats Should Be Provided, and the Quantity Is Important

The importance of seating has been highlighted in many studies, while how many seats should be provided has not been given enough attention. In particular, as they are closely linked with leisure and social activity [17], on-site seats play significant roles in public open spaces. The existence [74] and design [28] of seats may have an important effect on space usage.

In this investigation, the seat design of a space was characterised in terms of the seat–circumference ratio, which should be in the range between 33.815% and 72.500%. This in-

vestigation also discovered that, when there are too many seats and the seat–circumference ratio is higher than 72.500%, the environmental assessment of residents will be even lower than if few seats are provided (i.e., in the range of 0–33.815%).

*6.4. Preferred Environmental Features*

Based on the present analysis, spaces with green-vision rates between 20.385% and 46.990%, shading rates less than 93.275%, and seat–circumference ratios between 33.815% and 72.500% were most attractive to residents.

In addition, during practical design, when advantage value intervals of each indicator cannot be satisfied at the same time, it will be a dilemma for designers to prioritise between spatial features. In this context, this study provides a decision-making reference for all architects, following the path of "step 1: the green-vision rate should be greater than 20.385%; step 2: the shading rate should be less than 93.75%; step 3: the seat–circumference ratio should be less than 72.500%; and step 4: the seat–circumference ratio should be greater than 33.815%".

*6.5. More Trees or More Seats, Mutually Exclusive within Small-Scale Open Spaces*

As described in the previous section, the preference for a high green-vision rate is influenced by the distribution of seats. For spaces with a seat–circumference ratio less than 33.815%, sites with a green-vision rate higher than 41.080% are preferred; when the seat–circumference ratio is between 33.815% and 72.500%, the preferable green-vision rate will be in the range of 20.385% to 46.990%.

This indicates that spaces with low greenery and too many or too few seats will be less preferred. When these two indicators are in a relatively moderate range, spaces with more seats have a lower requirement for greenery; for spaces with fewer seats, more trees and greenery will be needed. This phenomenon was first identified in this investigation, and it is assumed to be due to people's sense of enclosure: both seats and surrounding greenery can provide the feeling of enclosure, and the sense of enclosure may have an optimal range for users; therefore, when a space is surrounded by more seats and feels more closed, more trees and plants will be less welcomed (and vice versa).

*6.6. Influence of Space Size and Pedestrians*

Based on the weight evaluation of the experts, site size and internal pedestrian flow may contribute significantly to the usage of a space and, among all indicators, space size was considered to be the most influential factor. Meanwhile, when users answered the question "do you feel this space worth frequently use in daily lives?", the importance of site size and internal pedestrian flow was found to be lower compared to that of the sky view factor, green-vision rate, and seat–circumference ratio.

On the one hand, results based on "worth use" are more related to the subjective preferences of the interviewees, but not the actual activity conducted; on the other hand, this result also revealed that to design small-scale open spaces within neighbourhoods, space size and internal pedestrian flow did not have significant impacts on environmental cognition and general assessment of a space. Therefore, when architects design open spaces in narrow residential areas, there is no need to feel too anxious: if a broad open space cannot be provided for residents, controlling the green-vision rate, sky view factor, and seat–circumference ratio may effectively improve environmental preferences for the space.

## 7. Limitations

*7.1. Skew of Space Size*

As shown in Figure 6, among all 104 data collection points, the distribution of the site area is relatively concentrated: the size of most spaces was concentrated between 0 and 25 square meters. This is mainly due to the fact that the measured spaces are located in high-density residential areas in Shanghai, which are characterised by a high building density and less spare land.

In decision tree analysis, extreme skew indicators will have low importance as artifacts. Among all of the selected indicators in this research, space size is the most skewed distribution; further research is needed to determine whether the importance of space size is as low as suggested by the results in affecting space usage.

Therefore, the importance of space size when designing public spaces in low-density residential areas or other contexts that can provide more options still needs to be reconfirmed. If spaces with a greater variety in size were observed, the importance could change considerably.

### 7.2. Subjective Self-Evaluations of Residents

In addition, the fact that the conclusions were derived from the subjective self-evaluations of residents, rather than an objective quantitative assessment, imposes limitations on the precision of the results. To answer the question of how space features affect environmental behaviour in small scale of open spaces, a more rigorous and objective investigation coupled with observational methodologies is needed.

### 7.3. Research Scope

For this study, measurements were obtained from 104 public spaces located in Yangpu District, Shanghai, China, as a data source. As such, the investigated spaces were relatively concentrated. On the one hand, this may help in the comparison of spatial differences in relatively similar environments, avoiding differences caused by environmental and socio-economic backgrounds and highlighting the influences of spatial features. On the other hand, this will lead to inevitable regional characteristics. Due to the urban building density of Shanghai, the investigated spaces were all in high-density residential areas. Compared to residential areas with lower density, the spaces in this area were relatively small in size. Therefore, whether this analysis result can be applied to residential areas in other district still requires further discussion.

## 8. Conclusions

This investigation endeavoured to establish a foundational understanding of the objective-driven elements and characteristics that influence the utilisation of open area spaces within neighbourhoods. The following two questions were examined: (1) Which environmental features significantly impact the public space of residential areas and should be incorporated into the analysis system? (2) How do relevant features influence environmental preferences within the analysis system?

First, an expert inquiry was first conducted to evaluate environmental features, and the analytic hierarchy process (AHP) was innovatively applied in the research field of environmental behaviour to determine the weight of each influencing factor. Based on this, key influencing indicators governing spatial utilisation were identified, forming the basis for the development of a relatively comprehensive evaluation index system. Based on this, the following was established:

- Five indicators, including the site size, internal pedestrian flow, sky view factor, green-vision rate, and seat–circumference ratio, were indicated to be the primary environmental features affecting outdoor space usage in the objective evaluation index.

Then, based on the first stage and indicators selected, on-site measurements were conducted across 104 spaces, accompanied by structured interviews with users of the spaces. Decision tree analysis was then employed to elucidate the decision-making processes of residents regarding their outdoor activities. The main findings are summarised as follows:

- Among these factors, the impacts of the sky view factor, green-vision rate, and seat–circumference ratio significantly outweighed those of the site size and internal pedestrian flow on the environmental preferences of residents.
- The use of a green-vision rate between 20.4% and 47.0%, a shading rate less than 93.3%, and a seat–circumference ratio between 33.8% and 72.5% can yield a better

space assessment. Moreover, a severe decline in preference can be anticipated when the green-vision rate falls below 20.4% or the shading rate exceeds 93.3%.

- The interaction between the green-vision rate and seat–circumference ratio affects the environmental preference of residents: spaces with more seats exhibit lower requirements for greenery, while spaces with fewer seats should prioritise trees and greenery.

Based on the above analysis, this research makes contributions of both academic and practical significance:

From an academic standpoint, this article proposes a relatively comprehensive index system comprising influencing factors pertinent to open spaces within neighbourhood areas. This index system facilitates a systematic understanding of the environmental characteristics affecting the usage of space.

At the practical level, this study offers valuable insights for future endeavours in neighbourhood design by analysing the decision-making mechanism of neighbourhood space users, as well as the importance of each considered indicator and the associated advantage value intervals.

**Author Contributions:** S.H.: Writing—Original draft, Methodology, Investigation, Data visualisation. D.S., F.S. and Y.Z.: Review and editing, Project administration, Conceptualisation. H.D. and M.Y.: Conceptualization, Writing—Original draft, Methodology, Supervision. All authors have read and agreed to the published version of the manuscript.

**Funding:** This work was supported by the National Natural Science Foundation of China [Grant No. 52078443 and No.52078231].

**Data Availability Statement:** Data are contained within the article.

**Conflicts of Interest:** The authors declare no conflicts of interest.

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
