# Peer review of "Assessing Neighbourhood Preference: An Evaluation of Environmental Features within Small-Scale Open Spaces"

_land, doi:10.3390/land13040531_

Round 1

Reviewer 1 Report (New Reviewer)

Comments and Suggestions for Authors

I have read the paper entitled “Assessing neighborhood preference: An evaluation of environmental features within small-scale open spaces”. Generally, the quality of paper is good.

The major comment is that the methodology section should be revised. How did the authors select indicators? Did the authors use experts’ opinions to select indicators?

The minor comment is that the authors did not provide research questions in the introduction section.

-        Research questions should be provided by the authors in the introduction section.

-        Methodology section is not clear. Are the authors selected indicators of their study using experts’ opinion? Fig. 1 indicates that the authors have used experts’ opinions in selecting indicators of study. But in the content of methodology, there is no evidence shows that authors have used experts’ opinions in the initial phase of research methodology.

-        The authors have invited 32 experts to fill in the questionnaires. Are these experts participated in selecting indicators? Why did the authors select 32 experts? Is the total sample 32? (I mean all experts agreed to participate)? If yes, what is the main advantage in your study that encouraged experts to participate?

-      The authors did mention the limitations of study. Importantly, limitations provide opportunities to other researchers to investigate new areas.

Author Response

Dear reviewer, thank you very much for taking the time to review the paper! We really appreciate your valuable comments. The feedback and modified paper are in the attechment. 

Reviewer 2 Report (New Reviewer)

Comments and Suggestions for Authors

The authors of this paper have done a lot of work. According to the characteristics of the specific space environment, the evaluation is carried out in combination with the site area, the flow of people, the green vision rate and other factors. In order to help this paper better published. I have some personal advice.

1. The current summary is not very good. According to the structure of research background, research purpose, research method, research process and research conclusion.

2. Why use AHP for research? Is this question a single factor influence analysis? Or a combination factor analysis? I think the author needs to explain. Why not use other methods such as QCA analysis method, grounded theory analysis method, etc. Because there needs to be a match between the research method and the research question. At the moment, I see no logical explanation.

3. The conclusion of the study is not summarized enough. The previous analysis of a lot, the process is very detailed, but the conclusion is too common, I did not see the use of the method after the innovation point. Or is the research question itself not innovative? If so, please make it clear in the first part of the paper what problem the topic addresses.

4. In Figure 12, is the role of this design illustrative? I think it needs to be supported by the views of other scholars. Why is it designed this way?

5. In Figure 4, please check the scale clearly. Without a compass, the mapping is not very regular.

6. Why choose YAAHP as the research tool? Instead of using the common SPSS software? Can't use it? Or what's the reason? Because YAAHP has no way to verify the reliability and validity of data. Please give a reasonable explanation.

7. I think the research on the total number is not very clear. What questions did the study focus on? What are the latest issues in this direction? In the literature review, I didn't see it!

8. In FIG. 5 and FIG. 6, detailed research records or certificates need to be supplemented. The basic information of the picture needs to be displayed, such as the data information such as the time and location of the picture. To make sure the research is real. The evaluation of the space is different in different seasons, and factors such as season and weather should be considered.

Therefore, I think the revised paper needs to be adjusted after the review.

Comments on the Quality of English Language

Minor editing of English language required

Author Response

Dear reviewer, thank you very much for taking the time to review the paper! We really appreciate your valuable comments. The feedback and modified paper are in the attechment. 

Reviewer 3 Report (New Reviewer)

Comments and Suggestions for Authors

The content of the article is valuable and beneficial. However, the article’s structure needs to be reviewed and modified to make its message clear to the reader. Dedicate the introduction only to the explanation of the research problem/gap (including questions and objectives) and the introduction of the article’s structure. Other theoretical topics should be presented in a separate section titled literature review, which results in the theoretical framework/conceptual model, including the index for evaluating environmental features within small-scale open spaces. Dedicate the third section to the study area and the fourth to the methodological details. The fifth, sixth, and seventh sections will be the results, discussion, and conclusions. Try to highlight the contribution of the research in the conclusion section. Providing suggestions (both practical and research) are also important.

Comments on the Quality of English Language

-

Author Response

Dear reviewer, thank you very much for taking the time to review the paper! We really appreciate your valuable comments. The feedback and modified paper are in the attechment. 

Reviewer 4 Report (New Reviewer)

Comments and Suggestions for Authors

The study presents a workflow for assessing small-space parks' attributes that contribute to the overall preferability of these spaces. The study is interesting and compiled interesting data. Analysis is rigourous.

Some key references are missing and are marked in the document. Also missing are discussions regarding the limitations of the approach in relation to the skewness of the greenspace size. The data seems to have parks of similar sizes, which is expected but the authors should bring this to the fore in evaluating the results.

Comments on the Quality of English Language

Language is acceptable, very few corrections required. Marked in text.

Author Response

Dear reviewer, thank you very much for taking the time to review the paper! We really appreciate your valuable comments. The feedback and modified paper are in the attechment. 

Round 2

Reviewer 3 Report (New Reviewer)

Comments and Suggestions for Authors

According to the authors’ responses and explanations and by comparing the new and previous versions of the manuscript, some concerns have been addressed, and the article’s message has become more precise. However, it is still suggested that the introduction and literature review sections be separated. In addition, introduce the case in a separate section with the title of the study area.

Author Response

Dear reviewer, thank you very much for taking the time to review the paper! We really appreciate your kindness help. The feedback and modified paper are in the attechment.

This manuscript is a resubmission of an earlier submission. The following is a list of the peer review reports and author responses from that submission.

Round 1

Reviewer 1 Report

Comments and Suggestions for Authors

Comments on the Quality of English Language

The manuscript requires English proofreading.

Reviewer 2 Report

Comments and Suggestions for Authors

The article is a comprehensive study aimed at understanding the factors influencing the usage of public open spaces in neighborhoods. The authors have employed a mix of methods including AHP analysis, expert inquiry, space measurement, on-site structured interviews, and decision tree analysis to assess various environmental indicators. The writing is relatively complete and the thinking is clear, the analysis methods in the article are quite innovative. There is the following suggestion or question:

(1) The introduction provides a good overview but could be enhanced by more directly stating the research gap the study aims to address.

(2) The justification for choosing specific methods, such as AHP analysis and decision tree analysis, could be more detailed.

(3) The results are comprehensive, but the presentation could be improved for clarity. Some tables and figures could be better integrated into the text with more detailed explanations to guide the reader through the findings.

(4) Discussing how the findings align or differ from previous studies would provide a deeper understanding of the study's contributions.

(5) The limitations are briefly mentioned, but a more thorough exploration of potential biases and the generalizability of the results would be beneficial. This could include discussing the study's applicability to different urban contexts.

Reviewer 3 Report

Comments and Suggestions for Authors

Assessing neighbourhood preference: An objective evaluation of public open spaces and their influential features

The research problem presented in this Paper is very interesting question and the topic of the great importance. As author/s said at the end of the Paper, and I agree, this research is important for both, practictioners and science. Initial idea, as well as theoretical foundation, methodology, results and discussion are adequate and sufficient.

Main Title depict what the research really represents. Abstract is informative enough; however, the structure should be slightly adjusted, indicating used methods. Keywords are well and informative enough.

Section Introduction is well conceived with a well-formed theoretical framework and elaborated approaches applied in other research studies.

Section Methods is very extensive and detailed, with clear organization line. In this part, author/s should give a study area explanation. It would be good to see some of the background participants characteristics which were interviewed. Please see Fig 1 and line 108, correction is needed, Figure 1 are doubled in this page. Figures 4,5,6 should be better incorporated into the text. The Results section is well presented and conceptualized. All graphic attachments are clear and understandable. In Section Discussion, authors made a good review of main conclusions and connect them with other results from similar researches, as a scientific confirmation of the obtained results. Please see Fig 11., doubled on page 17 and 18.

Section Limitation, could be incorporated into section Conslusions, which is the review of all presented results.

Reviewer 4 Report

Comments and Suggestions for Authors

The study explores factors influencing behaviour-environment interactions in neighborhood open areas, emphasizing their roles in outdoor participation and health promotion. Five primary indicators affecting outdoor space usage, with significant importance placed on the sky view factor, green-vision rate, and seat-circumference ratio are explored. The research proposes a composed metric  (influencing factors index, with a significance and weighting of indicators based on the AHP process) to better understand and inform future neighborhood design decisions.

The topic is timely and relevant, and it proposes interesting indicators to delve into. However, it explains very little, about methodological aspects and limitations of the indicators taken into account.

One significant drawback of this paper is the absence of a clear and defined explanation of the main indicators utilized. While the paper delves extensively into the AHP process, it overlooks a crucial aspect – the fundamental understanding and definition of the main indicators. Before delving into the pondering of indicators, it is essential to thoroughly comprehend what they represent, what explicitly they measure, and the basis on which they are established, including the methods used for data gathering and analysis.

At the moment this is very vaguely and deficiently described.

I suggest introducing a section in this paper with a clearer and expanded explanation of the basic 5 indicators and their definition and method of capturing.

The manuscript requires another thorough review and proofreading.

- There are many grammatical or text-related flaws in the manuscript such as, e.g.: line 100: “And based on indictors selected, decision tree can help to understand the inner mechanism that how do factors collaborative affect public space usage”

e.g. line 194: To get a precise understanding of roles specific size played in open spaces, 194 only area can be actual used were calculated, the area of roads, flower terrace, facilities, were excluded during the size calculating.”

-There are quite many sentences in the text that lack content strength, and are also grammatically weak: see e.g. line 114: “And it could also offer a valuable insight for future endeavours in neighbourhood design by analyzing the decision-making mechanism of neighbourhood space users, importance of each indicator, and their advantage value intervals”

- A figure appears in line 207 in the middle of the text;  no reason why is there – probably should be removed – it appears in another place later on in the paper.

Since many other similar flows appear in the manuscript, I consider this paper not mature for publishing in this phase.

Comments on the Quality of English Language

The study explores factors influencing behaviour-environment interactions in neighborhood open areas, emphasizing their roles in outdoor participation and health promotion. Five primary indicators affecting outdoor space usage, with significant importance placed on the sky view factor, green-vision rate, and seat-circumference ratio are explored. The research proposes a composed metric  (influencing factors index, with a significance and weighting of indicators based on the AHP process) to better understand and inform future neighborhood design decisions.

The topic is timely and relevant, and it proposes interesting indicators to delve into. However, it explains very little, about methodological aspects and limitations of the indicators taken into account.

One significant drawback of this paper is the absence of a clear and defined explanation of the main indicators utilized. While the paper delves extensively into the AHP process, it overlooks a crucial aspect – the fundamental understanding and definition of the main indicators. Before delving into the pondering of indicators, it is essential to thoroughly comprehend what they represent, what explicitly they measure, and the basis on which they are established, including the methods used for data gathering and analysis.

At the moment this is very vaguely and deficiently described.

I suggest introducing a section in this paper with a clearer and expanded explanation of the basic 5 indicators and their definition and method of capturing.

The manuscript requires another thorough review and proofreading.

- There are many grammatical or text-related flaws in the manuscript such as, e.g.: line 100: “And based on indictors selected, decision tree can help to understand the inner mechanism that how do factors collaborative affect public space usage”

e.g. line 194: To get a precise understanding of roles specific size played in open spaces, 194 only area can be actual used were calculated, the area of roads, flower terrace, facilities, were excluded during the size calculating.”

-There are quite many sentences in the text that lack content strength, and are also grammatically weak: see e.g. line 114: “And it could also offer a valuable insight for future endeavours in neighbourhood design by analyzing the decision-making mechanism of neighbourhood space users, importance of each indicator, and their advantage value intervals”

- A figure appears in line 207 in the middle of the text;  no reason why is there – probably should be removed – it appears in another place later on in the paper.

Since many other similar flows appear in the manuscript, I consider this paper not mature for publishing in this phase.
